# TAS: Personalized Text-guided Audio Spatialization

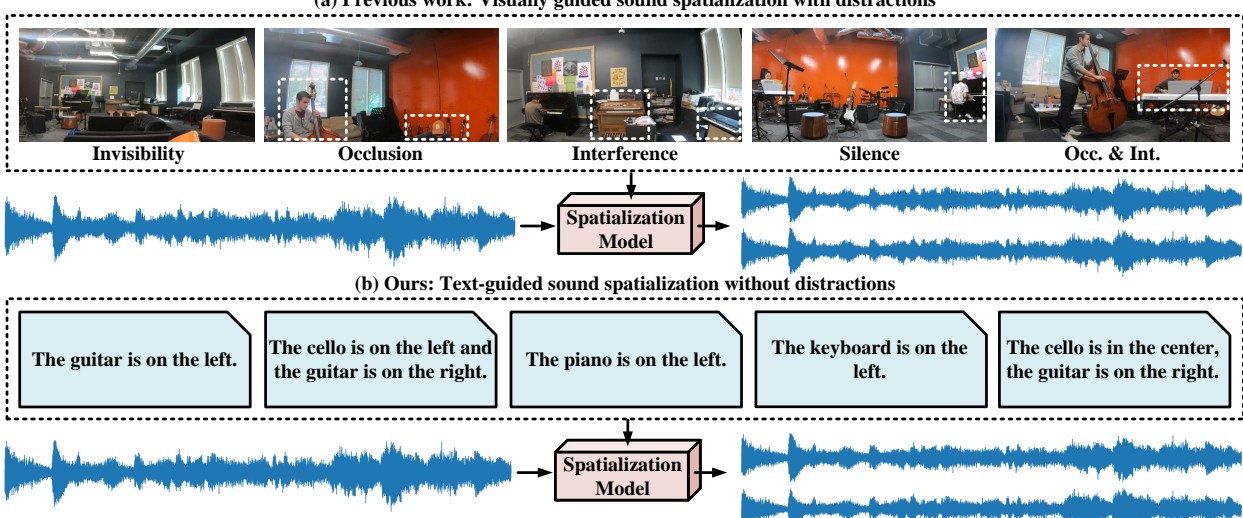

**Figure 1: We explore a new text-guided audio spatialization task that reconstructs the spatiality of audio based on text prompts. Compared with previous visually guided methods, it circumvents the pitfalls of visual guidance and allows for greater personalization. The text in (b) corresponds to the frames in (a).**

## ABSTRACT

Synthesizing binaural audio according to personalized requirements is crucial for building immersive artificial spaces. Previous methods employ visual modalities to guide the spatialization of audio because it can provide spatial information about objects. However, the paradigm is dependent on object visibility and strict audiovisual correspondence, which makes it tough to satisfy personalized requirements. In addition, the visual counterpart to the audio may be crippled or even non-existent, which greatly limits the development of the field. To this end, we advocate exploring a novel task known as Text-guided Audio Spatialization (TAS), in which the goal is to convert mono audio into spatial audio based on text prompts. This approach circumvents harsh audiovisual conditions and allows for more flexible individualization. To facilitate this research, we construct the first TASBench dataset. The dataset provides a dense frame-level description of the spatial location of sounding objects in audio, enabling fine-grained spatial control. Since text prompts contain multiple sounding objects and spatial locations, the core issue of TAS is to establish the mapping relationship between text semantic information and audio objects. To tackle this issue, we design a Semantic-Aware Fusion (SAF) module to capture text-aware audio features and propose a text-guided diffusion model to learn the spatialization of audio, which can generate spatial audio consistent with text prompts. Extensive experiments on TASBench compare the proposed method with several methods from related tasks, demonstrating that our method is promising to achieve the personalized generation of spatial sense of audio under text prompts.

## CCS CONCEPTS

• **Computing methodologies → Machine learning approaches**; **Artificial intelligence**.

## KEYWORDS

Text-guided generation, Audio spatialization, Audio synthesis, Multimodal learning

## 1 INTRODUCTION

Generating music, speech, and sound effects based on personalized requirements [3, 22, 39] is critical for applications such as augmented reality, virtual reality, game development, and video editing [6, 36]. In recent years, audio generation research has made surprising progress [13, 28, 30, 34]. However, existing studies mostly focus on the semantic generation of audio and ignore the significance of audio spatiality. Due to the binaural structure, human hearing conveys critical spatial information [4]. We can easily identify the

approximate location of a sound source and even track it without relying on vision.

An artificial head prosthesis records realistic binaural audio to provide users with a rich listening experience. However, binaural audio recording is expensive and relies on professional equipment [32]. Even though the device has been around for many years, binaural audio is still scarce. It is wise to eschew hardware and design algorithms to obtain binaural audio. Therefore, converting mono audio to binaural audio has attracted increasing attention [18, 20, 21]. Binaural audio with real human hearing experience can provide users with an immersive artificial 3D experience and has broad application prospects [1].

Recent work attempts to exploit visual modalities to synthesize binaural audio from mono audio [8, 20, 23, 38, 40]. Ideally, it works because the visual modality contains the spatial information of the objects required for audio spatial recovery. However, relying on visual modality requires two prerequisites: (i) The visual and audio modalities must be strictly time-synchronized; (ii) The sounding object must be visible. Object visibility guarantees may be contradictory. To include the object, the field of view needs to be as wide as possible. However, a larger field of view introduces more irrelevant objects and background while narrowing the object. In the case of audiovisual synchronization, we summarized 5 common situations of visual guidance, including *invisibility*, *occlusion*, *interference*, *silence* and their *combinations*, as shown in Figure 1 (a). Therefore, it is very challenging to exploit visual modalities to generate customized spatial audio.

The natural defects of visual guidance motivate us to search for a new way to synthesize spatial audio. Recently, text prompt-based generative methods have achieved impressive performance, such as audio style conversion [14, 17, 26], text-to-speech synthesis [2, 9, 31], etc. Generative models demonstrate powerful language understanding and generation performance. Inspired by this, we made the first attempt in this field to reconstruct the spatial sense of audio using text prompts, as shown in Figure 1 (b). Our approach circumvents the pitfalls of visual guidance and allows for greater personalization. In addition, it is capable of serving as a promising remedy when visual guidance methods fail.

In this paper, we raise the issue of text-guided audio spatialization, which requires models to generate spatially consistent spatial audio from mono audio based on text prompts. To facilitate this research, we present the TASBench dataset, the first dense frame-level annotated spatial audio dataset that provides textual annotations for the spatial locations of sounding objects. The text prompt contains multiple sounding objects and different spatial locations. To connect the semantic information in the text prompts with the sounding objects in the audio, we design a Semantic-Aware Fusion (SAF) module to capture text-aware audio features. Then, we propose a new text-guided audio spatialization method as a first baseline for the TAS task. The method follows the probabilistic diffusion model structure, using mono waveforms and text prompts as guiding conditions to generate differential waveforms. We conduct extensive and rich experiments to verify that the proposed method and module are practicable and effective. Experimental results show that it is prospective to convert mono audio into spatial audio using text prompts. In general, the contributions of the paper can be summarized as follows:

1) We explore a novel text-guided audio spatialization task, and introduce TASBench, a new dataset that provides dense frame-level spatial location annotations of sounding objects for TAS.

2) We design a semantic-aware fusion module to establish the perceptual relationship between text and audio modalities and propose a text-guided diffusion model for spatial audio generation.

3) Extensive experimental results demonstrate the feasibility and potential of our text-guided audio spatialization method. Comparisons with several methods from related tasks demonstrate the superiority of the proposed method.

## 2 RELATED WORK

### 2.1 Text-Guided Audio Generation

By integrating text and audio, we can mine diverse audio information and expand the use scenario of audio [16]. Recently, with the vigorous growth of deep generative models, text-guided generation has been intensively investigated in the audio field [29]. A prevalent text-guided generation task is audio style conversion, which seeks to convert only specific attributes while leaving other properties unaltered, such as speech conversion [14, 16], emotion conversion [10, 41], etc. Le *et al.* [16] proposed a large-scale speech-text guided generation model. It is based on non-autoregression and can achieve zero-shot style conversion within a single or cross-language. Guo *et al.* [10] proposed an emotion-controllable speech generation model in which the intensity of emotion is guided by a soft label. Another common task is text-to-speech synthesis, which attempts to synthesize a target audio-style speech within a given text. Kreuk *et al.* [15] proposed an autoregressive audio generation model to generate high-quality audio under conditional and unconditional settings. However, it is difficult for autoregressive methods to achieve fine-grained control of audio properties. Huang *et al.* [12] proposed a diffusion model with prompt enhancement, which utilizes latent diffusion mechanisms and autoencoders to generate audio. Yang *et al.* [35] explored text-to-audio synthesis based on a discrete diffusion process that handles audio codes in VQ-VAE [33], employing masked text generation with CLIP [24] representation. To solve the quality problem caused by quantization, Shen *et al.* [29] proposed an audio codec based on a residual vector quantizer, and then utilized a diffusion model to synthesize latent vectors based on text prompts. Huang *et al.* [11] proposed a diffusion probabilistic method for generating music from text prompts that outperformed methods conditioned on simple music attributes. Currently, most text-guided audio generation mostly focuses on the mono audio level and ignores the spatial properties of audio.

### 2.2 Audio Spatialization

Reconstructing the spatial properties of sound from mono audio has received long-standing attention. Traditional methods utilize linear mapping relationships to convert mono audio to spatial audio, making it difficult to simulate realistic non-linear scenes [7, 37]. Therefore, synthesizing spatial audio using the nonlinear modeling capabilities of deep learning has received widespread attention [38, 40]. Due to the strong correlation between vision and hearing, it is natural to associate spatial audio generation with visual modality. Visually guided audio spatialization has been extensively studied in

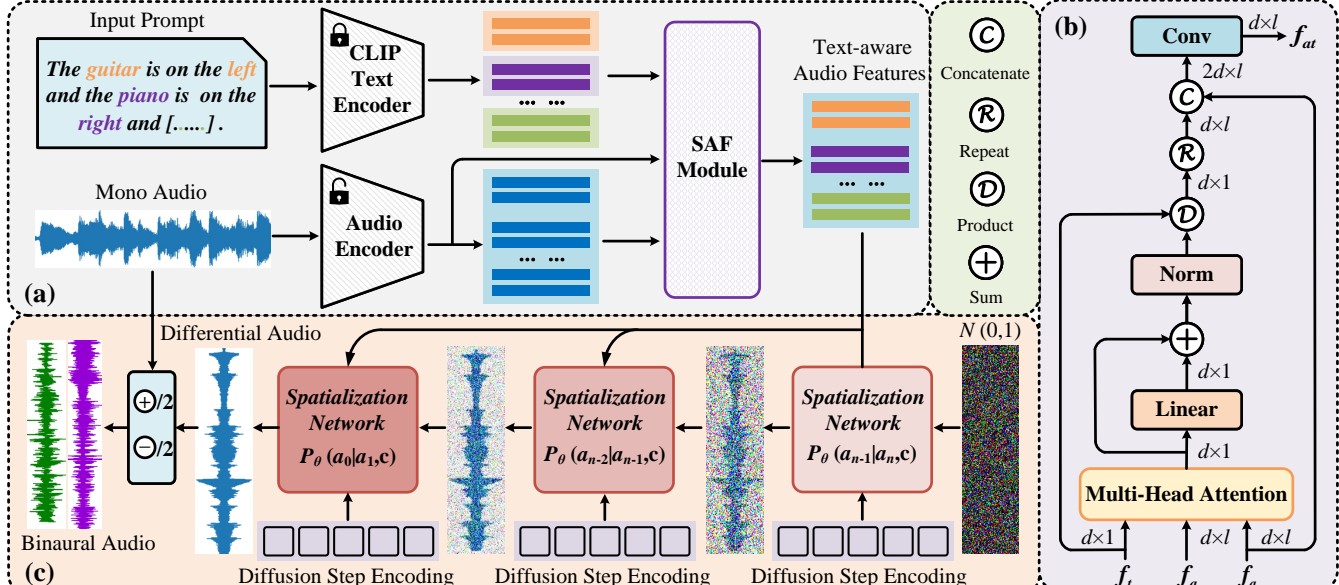

**Figure 2: Illustration of our text-guided audio spatialization method. In (a), text prompts and audio are fed into the encoder to extract features and then fused through the SAF module. In (b), the SAF module establishes the semantic perception relationship between text and audio. In (c), the aggregated text-aware audio features provide guidance information to the spatialization model to support the reconstruction of audio spatial perception.**

recent years [8, 18]. In the pioneering approach, Gao *et al.* [8] proposed a visually guided UNet-structured codec to generate binaural audio via spectrograms. Next, Zhao *et al.* [40] introduced sound source separation into binaural audio generation. The method improves spatialization performance by incorporating both tasks into a unified training framework through a shared visual network. However, this paradigm makes binaural audio generation subject to additional source-separated datasets and models. Li *et al.* [18] proposed a generative adversarial approach to synthesize binaural audio. The visual modality provides guidance information to the generator while providing visual reference to the discriminator.

Although these existing audio spatialization methods can generate spatial audio that corresponds to visual content, they rely on strict audiovisual correspondence and have limited flexibility. Different from the above methods, we explore a new task of text-guided audio spatialization and propose a text-guided diffusion model to reconstruct the spatial sense of audio. Experimental results demonstrate that our approach can generate spatial audio that is consistent with text prompts and outperforms some visually guided methods.

## 3 METHOD

To address the TAS issue, we present a text-guided audio spatialization method based on the diffusion model to realize audio spatialization reasoning guided by text prompts. Our method recovers differential audio from random noise conditioned on mono audio and text prompts. Mono audio provides the original signal basis, while text prompts provide spatial information during generation. Figure 2 shows the pipeline illustration of the proposed method.

### 3.1 Problem Formulation

Given a mono audio $A_m$ and a text prompt $T$, our model aims to synthesize the binaural audio $A_b = \{A_l, A_r\}$ corresponding to the text prompt. $A_m$ can be obtained by mixing channels of binaural audio $A_b$:

$$A_m = A_l + A_r. \tag{1}$$

For simplicity, instead of directly synthesizing the binaural audio itself, we predict the differential audio $A_d$ of the binaural audio:

$$A_d = A_l - A_r. \tag{2}$$

To convert the differential audio waveform distribution $q(A_d)$ into a Gaussian noise distribution $\mathcal{N}(0, \mathbf{I})$, we diffuse it via $N$ steps in the diffusion process. $a$ is used to represent $A_d$ for convenience. Then, the diffusion process can be described as:

$$q(a_{1:N}|a_0) := \prod_{n=1}^{N} q(a_n|a_{n-1}). \tag{3}$$

In the reverse process, our diffusion model with parameter $\theta$ generates clean differential audio from Gaussian noise $p(x_T) \sim \mathcal{N}(0, \mathbf{I})$ guided by both mono audio $A_m$ and text prompts $T$. The reverse process can be expressed as:

$$p_\theta(a_{0:N}) := p(a_N) \prod_{n=1}^{N} p_\theta(a_{n-1}|a_n, A_m, T). \tag{4}$$

Finally, the generated differential audio $A_d'$ is computed with the mixed audio $A_m$ to obtain the final binaural audio $A_b' = \{A_l', A_r'\}$:

$$A_l' = \frac{A_m + A_d'}{2}, A_r' = \frac{A_m - A_d'}{2}. \tag{5}$$

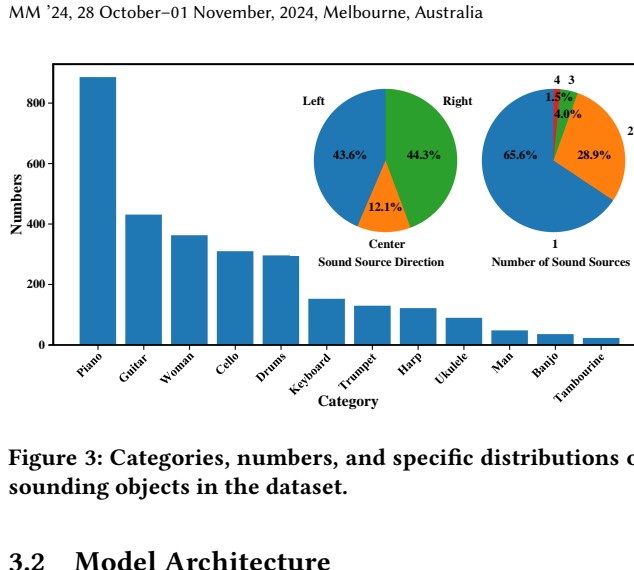

Figure 3: Categories, numbers, and specific distributions of sounding objects in the dataset.

## 3.2 Model Architecture

**Audio Encoder:** The audio encoder takes mono audio as input and outputs the extracted audio features in the form of waveforms. The audio encoder consists of two 1D convolutional layers to extract audio features $f_a = [f_a^1, \ldots, f_a^L] \in \mathbb{R}^{d \times L}$, where $d = 512$ is the embedding dimension and $L = 16000$ is the audio clip length. The weights of the audio encoder are not frozen during training.

**Text Encoder:** For input prompts, the text encoder of pre-trained CLIP [24] is employed to extract text features $f_t \in \mathbb{R}^{d \times 1}$, where $d = 512$ is the embedding dimension. We also utilize diverse text encoders to extract text features to verify the impact of text representation on audio spatialization, including BERT [5] and T5 [25] (introduced in Sec. 4.6). We perform average pooling on the embedding representations of all tokens to obtain the information of the entire text prompt in BERT and T5. A learnable fully connected layer is used to reduce the feature dimension of BERT from 768 to 512. Following the common practice, the weights of all text encoders are frozen during training.

**Semantic-Aware Fusion Module:** To highlight key audio semantic information closely related to text prompts, we propose a semantic-aware fusion module. Specifically, given text features $f_t$ and audio features $f_a$, the semantic-aware fusion module learns to aggregate text-aware audio features $f_a'$. The text-aware audio features can be computed as:

$$f_a' = \sum_{l=1}^{L} w_l^a f_a^l = \sigma(\frac{f_t f_a^{\mathsf{T}}}{\sqrt{d}}) f_a. \tag{6}$$

Then, the text features are fused with text-aware audio features:

$$f_{at}' = f_t \odot (\text{FC}(f_a') + f_a'), \tag{7}$$

where $\odot$ represents the matrix dot product, and FC represents the fully connected layer. Finally, $f_{at}'$ is repeated $L$ times to splice with audio features and fused through a convolutional layer to obtain the final aggregated text-aware audio features $f_{at}$:

$$f_{at} = \text{Conv}(\text{Concat}[f_{at}', f_a]), \tag{8}$$

where Conv represents a 1D convolution layer, and Concat represents element-wise concatenation.

**Spatialization Network:** The spatialization network consists of 3 residual blocks, and each residual block has 10 dilated 1D convolutional layers. The diffusion step is encoded using two fully

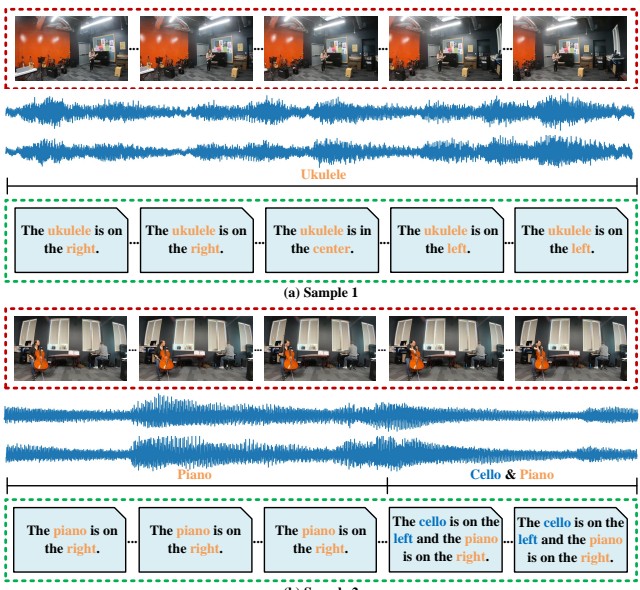

Figure 4: Examples of the TASBench dataset. We implement dense frame-level annotation of the sounding state and spatial location of sounding objects.

connected layers. Next, the proposed semantic-aware fusion module is used to highlight text-aware audio features and fuse audio and text features. Then, the aggregated text-aware audio features are injected into each residual block to provide guidance information. Finally, the ultimate output of the model is the concatenation of the outputs of all residual blocks.

## 4 EXPERIMENTS

We first introduce the construction of TASBench, implementation details, and evaluation metrics in this section. Then, we conduct comparative experiments between the proposed baseline and methods from related tasks. We also provide experimental results of several task-related methods guided by text prompts. In addition, we compare the impact of different text representations on audio spatialization and verify the personalization generation capability of the proposed method. Finally, we analyze the proposed method in ablation experiments and user study.

### 4.1 TASBench

**Dataset Statistics:** We build the first text-guided audio spatialization dataset to facilitate related research. A straightforward idea is to provide only one text annotation for an audio clip. In contrast, we provide dense spatial location annotations for sounding objects in spatial audio, which allows us to engage in fine-grained control of spatial sense generation. All spatial audio is collected from the FAIR-Play dataset [8], which was recorded in a music room with professional equipment. The TASBench dataset contains 1871 10-second audio clips and 190625 text annotations over 12 categories, covering sounds from musical instruments, humans, and their various combinations. Figure 3 shows the specific categories

| Modality | Method | STFT↓ | ENV↓ | WAV↓ | AMP↓ | PHA↓ | SNR↑ |
|---|---|---|---|---|---|---|---|
| w/o Modality | Mono-Mono | 1.155 | 0.153 | 7.666 | 0.267 | 0.592 | 5.735 |
| Visual Modality | L2BNet [23] | 1.028 | 0.148 | - | - | - | - |
| | MONO2BINAURAL [8] | 0.959 | 0.141 | 6.496 | 0.252 | 0.591 | 6.232 |
| | APNet [40] | 0.889 | 0.136 | 5.758 | 0.247 | 0.585 | 6.972 |
| | Sep-stereo [40] | 0.879 | 0.135 | 6.526 | 0.256 | 0.590 | 6.422 |
| | Main network [38] | 0.867 | 0.135 | 5.750 | 0.246 | 0.583 | 6.985 |
| | Complete network [38] | 0.856 | 0.134 | 5.787 | 0.247 | 0.584 | 6.959 |
| | SAGM [18] | 0.851 | 0.134 | 5.684 | 0.243 | 0.570 | 7.044 |
| | AVSN [20] | 0.849 | 0.133 | - | - | - | - |
| Text Modality | MONO2BINAURAL [8] | 0.980 | 0.143 | 6.534 | 0.252 | 0.593 | 6.223 |
| | APNet [40] | 1.003 | 0.144 | 6.686 | 0.254 | 0.593 | 6.128 |
| | Ours | 0.945 | 0.140 | 6.230 | 0.249 | 0.591 | 6.613 |
| | Ours+SAF | 0.914 | 0.137 | 6.092 | 0.245 | 0.586 | 6.771 |

**Table 1: Comparison with methods from related tasks. Modality refers to the guidance that provides spatial information for the audio spatialization model.**

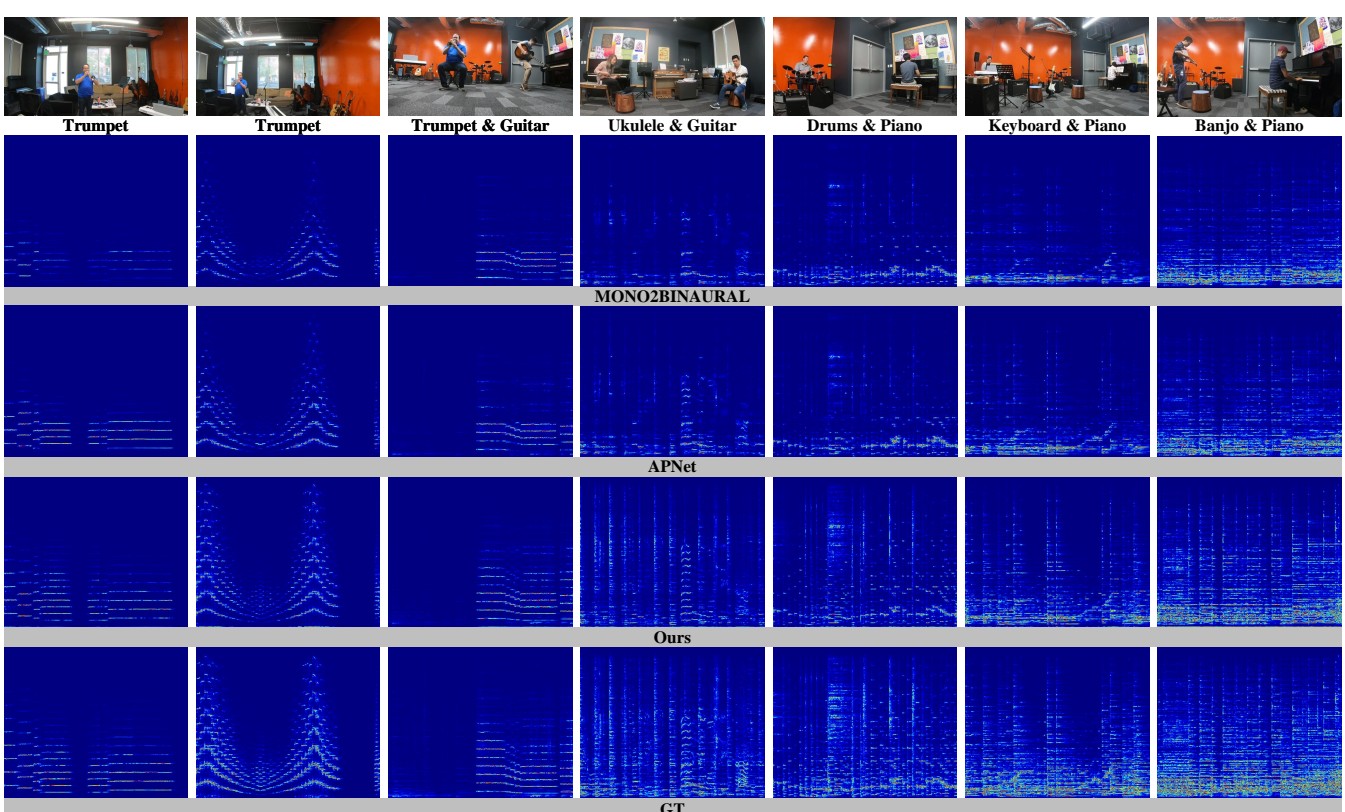

**Figure 5: Visualization results of differential spectrogram generated by our approach and task-related methods. MONO2BINAURAL and APNet are based on visual guidance. GT represents the ground truth.**

and the number of each category. There may be multiple categories in an audio clip, so the total number of categories exceeds the total number of audios. The number of sound source categories in an audio clip ranges from 1 to 4. Their proportions are shown in the right pie chart in Figure 3. In the left pie chart, we also compute the distribution of sound source directions. Overall, the distribution of the TASBench dataset is reasonable.

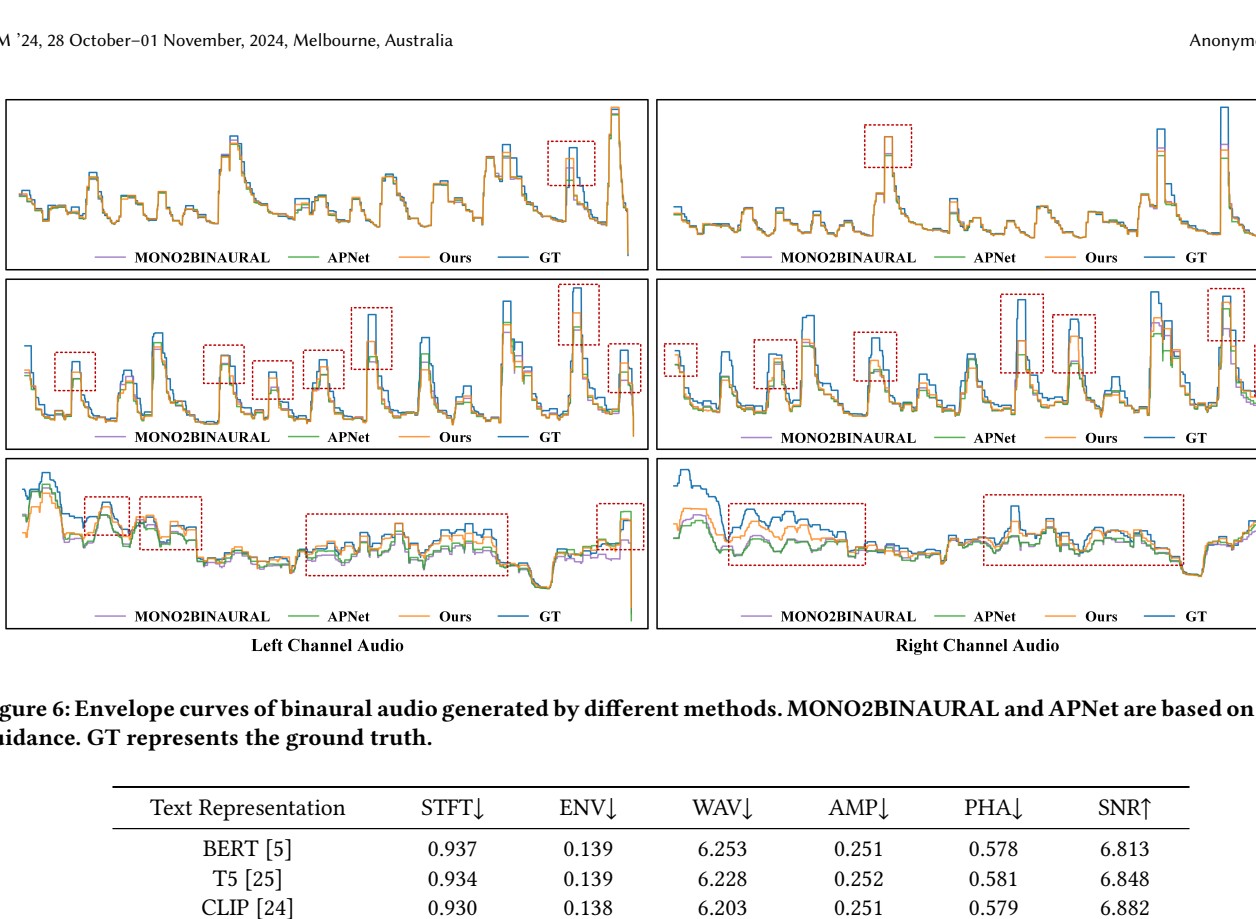

**Figure 6: Envelope curves of binaural audio generated by different methods. MONO2BINAURAL and APNet are based on visual guidance. GT represents the ground truth.**

| Text Representation | STFT↓ | ENV↓ | WAV↓ | AMP↓ | PHA↓ | SNR↑ |
|---|---|---|---|---|---|---|
| BERT [5] | 0.937 | 0.139 | 6.253 | 0.251 | 0.578 | 6.813 |
| T5 [25] | 0.934 | 0.139 | 6.228 | 0.252 | 0.581 | 6.848 |
| CLIP [24] | 0.930 | 0.138 | 6.203 | 0.251 | 0.579 | 6.882 |

**Table 2: Evaluation of text representations across different encoders on the split1 segmentation of TASBench dataset.**

**Annotation:** To better annotate the spatial location of sounding objects, we combine spatial audio and video provided by the FAIR-Play dataset. Specifically, we first extract visual images from each video at 10 frames per second. We then provide a spatial location annotation for each sounding object in each image frame based on its visual and auditory spatial location. There are some special cases in the annotation process: (i) Audio spatial perception may not exactly match the visual situation when the sounding object is visible. In this case, we utilize visual information as the main annotation to faithfully reflect the spatial position of the sounding objects. (ii) We utilize spatial audio as the main annotation object when the sounding object is invisible. Since the FAIR-Play dataset lacks specific spatial location records, the spatial location of the sounding object can be roughly divided into three directions: Left, Center, and Right. We use text to annotate the spatial location and status of the sounding objects in order from left to right. We only annotate sounding objects. When the object is completely silent, it will not appear in our annotations. Figure 4 shows some samples of the constructed dataset.

## 4.2 Implementation Details

**Dataset Settings:** We train and test the proposed method and other methods from related tasks on the constructed TASBench dataset. We retain the original partition of the dataset, which is 1497/187/187

for training, validation and testing. The average of 10-fold cross-validation of the model is used as the final result. We randomly intercept a 1-second clip from a 10-second audio sample with a sampling rate of 16kHz. The text closest to the timestamp in the center of the clip is chosen as the input prompt.

**Training and Inference:** The batchsize of the model is 12 and the maximum number of training steps is 3000. The proposed model is optimized using Adam with a learning rate of 2e-4 during training. We utilize a window with a hop size of 0.1 seconds to generate audio during inference.

## 4.3 Metrics and Methods from Related Tasks

**Metrics:** Six evaluation metrics are employed to comprehensively evaluate generated binaural audio, including STFT Distance [8], Envelope (ENV) Distance [19], Wave L2 (WAV×$10^{-3}$) [27], Amplitude L2 (AMP), Phase L2 (PHA), and Signal-to-Noise Ratio (SNR) [21]. See supp. for details.

**Methods from Related Task:** We compare the proposed method with several visually guided methods from related tasks, including weakly semi-supervised method: L2BNet[23]; autoencoder-based methods: MONO2BINAURAL [8], AVSN [20]; multi-tasking-based methods: APNet [40], Sep-stereo [40]; attention-based methods: Main network [38], Complete network [38]; GAN-based method:

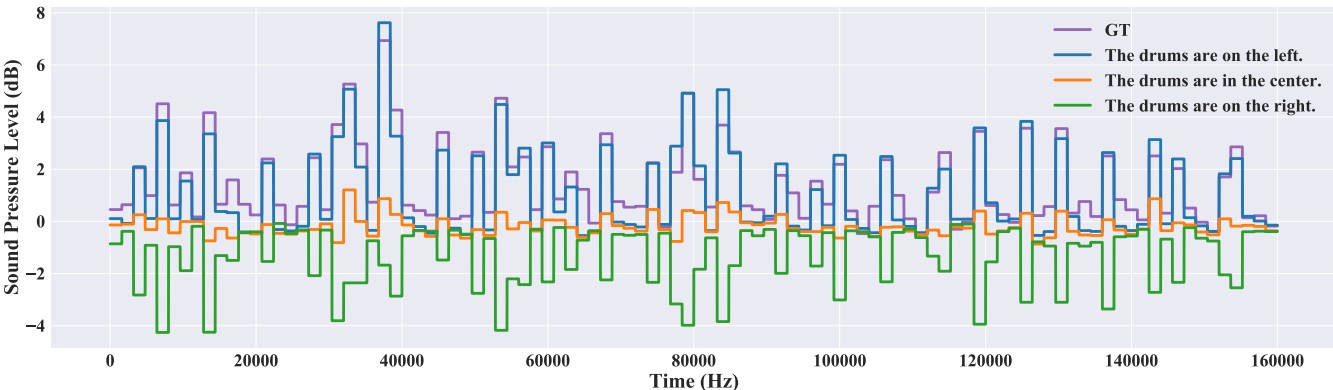

**Figure 7: The sound pressure level curve of the audio generated by our method under text prompts with different spatial senses. Left: SPL > 0dB, Center: SPL = 0dB, and Right: SPL < 0dB.**

| Modality | Fusion | STFT↓ | ENV↓ | WAV↓ | AMP↓ | PHA↓ | SNR↑ |
|----------|--------|-------|------|------|------|------|------|
| w/o Modality | - | 1.195 | 0.156 | 7.953 | 0.275 | 0.588 | 5.693 |
| Audio-Only | - | 1.078 | 0.148 | 7.195 | 0.265 | 0.602 | 6.090 |
| Audio-Text | Add | 0.967 | 0.141 | 6.446 | 0.255 | 0.585 | 6.731 |
| Audio-Text | Conv | 0.959 | 0.141 | 6.402 | 0.253 | 0.586 | 6.738 |
| Audio-Text | SAF | 0.930 | 0.138 | 6.203 | 0.251 | 0.579 | 6.882 |

**Table 3: The ablation results of our approach on the split1 segmentation of TASBench dataset.**

SAGM [18]. Mono-Mono represents fake binaural audio created by channel duplication of mono audio.

## 4.4 Quantitative Results

To investigate the impact of different guidance modalities on audio spatialization and prove the superiority of our method, we organize extensive quantitative experiments. Table 1 demonstrates the quantitative results under different guidance. Mono-Mono is audio without modal guidance, which has no spatial awareness. The improved evaluation metrics show that both our method and other methods can generate audio with spatial awareness. As can be seen, our method even outperforms some visually guided methods. This is challenging because most of the visual images in the FAIR-Play dataset have clear visual expressions.

For intuitive comparison, we replaced the visual modality in MONO2BINAURAL and APNet with text prompts and retrained it. Some metrics decreased in text guidance compared to visual guidance in the MONO2BINAURAL method, which further supports our observations. In the text-guided method, Ours represents the baseline model without the SAF module, and Ours+SAF represents the baseline model with the SAF module. After eliminating the influence of modality, our method outperforms other methods in all metrics, indicating the superiority of the proposed model. The proposed SAF module further significantly improves the performance of our method by incorporating text-aware audio features. This demonstrates the effectiveness of the proposed module.

## 4.5 Qualitative Results

We visualize the differential spectrogram generated by different methods in Figure 5. The first row is the visual image corresponding to the text prompt. The second and third rows are the results generated by the visually guided method, while the fourth row is the result generated by our text-guided method. It can be seen that the time-frequency structure of the spectrogram generated by our method is comparable to the ground truth.

To visually observe the similarity of the generated waveforms, we visualize the envelope curves of the left and right waveforms, as shown in Figure 6. The warping of the waveform envelope curve generated by the proposed method follows the ground truth situation well. Compared with MONO2BINAURAL and APNet, the proposed method has better warpage peak, and the waveform shape is closer to the ground truth. Overall, our method demonstrates competitive performance with visually guided generation methods in both spectrogram and waveform domains. This demonstrates the promise of using text prompts to synthesize mono audio into binaural audio.

## 4.6 Text Representation

We use existing text encoders for feature extraction on text prompts, including CLIP, BERT, and T5. CLIP is a pre-trained model obtained through contrastive learning of multimodal data and is widely used in text feature extraction. BERT is a large language model pre-trained on a large amount of plain text corpus. Compared with BERT, T5 is a more versatile large language model that can adapt to a

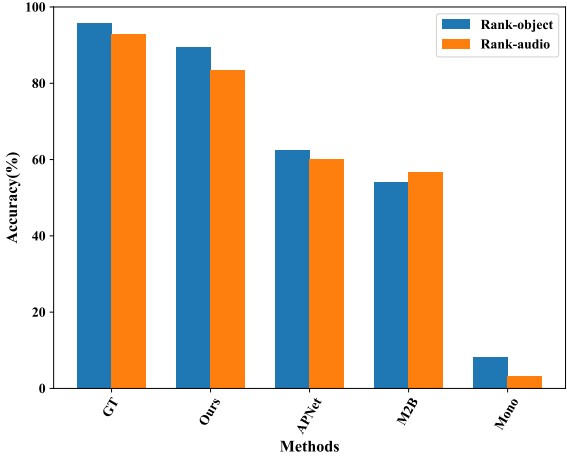

**Figure 8: User study of direction perception. All samples are generated based on text prompts. Mono and M2B represent Mono-Mono and MONO2BINAURAL, respectively.**

variety of natural language processing scenarios. Table 2 shows the quantitative results of binaural audio generated based on different text representations. It can be seen that the text representation of CLIP is superior to other methods in most metrics, demonstrating its potential in audio spatialization tasks.

### 4.7 Personalized Results

To evaluate the personalization capability of the proposed method, we conduct personalization experiments. Specifically, given a mono audio, we then feed it to our model with text prompts of different spatial senses. Finally, the difference curve of the sound pressure level is employed to visualize the spatial variations of the audio. The SPL represents the difference in sound pressure level between the left channel and the right channel of audio. The spatial sense is biased to the left when SPL > 0, to the right when SPL < 0, and toward the center when SPL is near 0. The personalization results of our method are demonstrated in Figure 7. The purple line is the visualization result of the ground truth, and the sense of space is biased to the left. It can be seen that the spatial sense of the audio generated by our method is comparable to the ground truth (blue line). When the model is provided with different text prompts (orange and green lines), it can generate audio consistent with the text prompts, demonstrating the personalization capabilities of the proposed method.

### 4.8 Ablation Results

We implement ablation experiments to demonstrate the impact of text modality and fusion methods on audio spatialization. Table 3 shows the results of the ablation experiments. The w/o Modality represents Mono-Mono, which is obtained by channel duplication of mono audio, and it has no spatial information. Audio-Only means using only mono audio as guidance without text prompts. Interestingly, most metrics of Audio-Only still improve without spatial information guidance. The reason may be that the model can learn some audio with insignificant spatiality, which does not need to

rely on explicit spatial information, such as audio with the object in the center.

The bottom of Table 3 shows the results of audio-text joint guidance. It can be observed that the introduction of text modality further boosts the performance of the model because it allows the model to generate audio corresponding to the spatial information in the text prompt. In addition, we also compare different fusion methods of text and audio modalities, including element-wise addition, 1D convolution and the proposed SAF module. Compared to these methods, our method achieves leading performance through perceptual interaction between text and audio modalities.

### 4.9 User Study

We conduct a user study to artificially estimate binaural audio generated by different methods. The user study recruited 8 subjects. We provide each subject with 30 sets of samples. Each set of samples contains audio generated by Mono-Mono, MONO2BINAURAL, APNet, and Ours, as well as the ground truth. The samples in each group are shuffled and named with random numbers. To ensure generalizability, all subjects were novices and were asked to give corresponding answers based on the audio and questions provided. Specifically, we use text prompts with location information removed as questions for each group of samples. Then, each subject was asked to listen to each audio one by one and given the spatial location of the sounding object, including left, center, and right. Finally, we compute the average accuracy of all participants under diverse methods as the final experimental results.

In Figure 8, we provide two levels of accuracy for user study, including rank-object and rank-audio. Rank-object evaluates the position accuracy of the sounding object from the object level. Rank-audio evaluates the accuracy of spatial perception from the entire audio level, *i.e.*, the positions of all objects need to be consistent with the ground truth. It can be seen that our approach can generate credible spatial audio, demonstrating the effectiveness of using text prompts to guide audio spatialization. Compared with other methods, the spatial audio generated by the proposed method is more realistic.

## 5 CONCLUSION

In this paper, we explore a novel text-guided audio spatialization task, which aims to convert mono audio into spatial audio based on text prompts. To promote research in this area, we construct the first dense text-annotated spatial audio dataset, named TASBench. We design a semantic-aware fusion module to capture text-aware audio features to enhance the correlation between text and audio features and propose a new text-guided audio spatialization model as a first baseline for this task. We compare the proposed method with several visually guided methods from related tasks and demonstrate that the proposed method can establish a connection between text prompts and the spatial perception of audio. Personalization experiments demonstrate that the proposed method can generate audio with a specified spatial sense based on text prompts. In the future work, we believe this research will facilitate multimodal audio spatialization, *i.e.*, generation under the joint guidance of text, audio, and visual modalities.

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
