# OpenReview forum: "TAS: Personalized Text-guided Audio Spatialization"
_acmmm.org/ACMMM/2024/Conference — MM2024 Poster_

### Official Review · Reviewer_AwGn · 2024-05-15

**Rating:** 1
**Confidence:** 4

**Summary:**

Text-guided audio spatialization (TAS) is indeed a novel idea, however, the manuscript still lacks much.

**Strengths:**

As mentioned in the manuscript, this work advocates exploring a novel task called text-guided audio spatialization (TAS), which aims to convert mono audio into spatial audio based on text prompts. This is a change from the previous fixed idea of using visual modalities to guide the audio spatialization. In order to build the mapping relationship between text semantic information and audio objects, a semantic-aware fusion (SAF) module is designed in the manuscript to capture text-aware audio features and learn the audio spatialization using the diffusion model, which is also different from the previous practice of using convolutional networks for audio spatialization. In addition, this work creates a TASBench dataset that provides a dense frame-level description of the spatial location of sounding
objects in audio, providing data support for text-guided audio spatialization.

**Limitations:**

Firstly, in the description of the method:

Q1: The emphasis of the method presentation is not satisfactory. Some common knowledge aspects of audio spatialization, such as the processing of the input audio and the principles of the diffusion model, are over-introduced, while the proposed text-guided audio spatialization method is only briefly introduced. In particular, the description of the Semantic-Aware Fusion (SAF) module, which is summarized as one of the innovations in the contribution.
(1)	Equations (6)-(8): Missing description of the parameters involved in the equations.
(2)	The parameters labeled in Figure 2(b) cannot correspond to Equations (6)-(8) in the manuscript. Is it true that Figure 2(b) has two parameters, the letter l and the number 1, or has it been plotted incorrectly?

Q2: Spatialization Network: The diffusion process as well as the output of the SAF are redundantly described without presenting a clear presentation of how this spatialized network handles these two inputs? What is the relationship between the two inputs? Sample concat or other processing?

Secondly, for the new dataset TASBench:

Q3: Except for the expanded categories of “Woman” and “Man”, the remaining category names are not fully aligned to the FAIR-Play dataset. Also, the FAIR-Play dataset involves some non-common instruments or non-instruments, such as wooden boxes that can be struck and bell-like objects that can be shaken to make them sound; how are these types of videos labeled?

Q4: What is the role of the statistics on the number of sound sources mentioned in Figure 3 in the process of training or testing? Which aspect does the phrase “Overall, the distribution of the TASBench dataset is reasonable.” (Section 4.1 Dataset Statistics) mean?

Q5: Section 1 mentions: “In the case of audiovisual synchronization, we summarized 5 common situations of visual guidance, including invisibility, occlusion, interference, silence and their combinations, as shown in Figure 1 (a)” However, the Section 4.1 does not explicitly describe how the dataset solves these problems. For example, how does it avoid interference from identical instruments? (e.g., the presence of guitars around guitars as well.)

Q6: Is the annotation manual or model-based? If it is model labeling, are there any metrics to judge whether the labeling is correct or not?

Thirdly, for the experiment:

Q7: The four metrics commonly used for binaural audio are STFT,ENV, Mag, and SNR; the manuscript and supplemental materials do not explain the reasons for not using the metric Mag, and the reasons for choosing a few other new metrics (which don't seem to have much to do with the use of text or visuals)

Q8: Comparisons of binaural audio performance using text-guided instead of vision-guided could not demonstrate superiority. As shown in Table 1, some methods, e.g.[8][40], the performance instead decreases after using the text instead of the visual. The manuscript explains that it is to highlight that text guidance has some effect, but obviously does not show the level of performance of the model proposed in this work.

Some additional details:

Q9: The layout of figures and tables. A figure or table is far away from the corresponding text description. They should be repositioned to avoid page breaks.

Q10: “Dataset Settings: … We retain the original partition of the dataset,” It has been pointed out in Paper 《Visually Informed Binaural Audio Generation without Binaural Audios》 that there are some problems with the division provided by the FAIR-Play dataset. It is recommended that the new division provided in Paper 《Visually Informed Binaural Audio Generation without Binaural Audios》 be used.

Q11: Does the differential spectrogram provided in Figure 5 take into account that the generated left and right channels are reversed? In addition, the spectrogram should be plotted with the appropriate coordinate system.

**Suitability:**

3

---

### Official Review · Reviewer_YELE · 2024-05-23

**Rating:** 3
**Confidence:** 2

**Summary:**

The paper presents TAS, a novel framework for text-guided audio spatialization, aiming to convert mono audio into spatial audio based on text prompts. Unlike traditional methods that rely on visual modalities for spatial information, TAS leverages textual descriptions to achieve greater flexibility and personalization in audio spatialization. The authors introduce TASBench, a new dataset with dense frame-level annotations of spatial audio, which enables fine-grained spatial control. The core contributions include a Semantic-Aware Fusion (SAF) module to integrate text and audio features and a text-guided diffusion model to generate spatial audio consistent with the text prompts. The method is evaluated extensively on the TASBench dataset, showing promising results compared to existing approaches.

**Strengths:**

1. Novel Approach: The shift from visually guided to text-guided audio spatialization is innovative. This approach circumvents the limitations of visual guidance, such as dependency on object visibility and strict audiovisual correspondence.
2. Dataset Contribution: The introduction of TASBench, a densely annotated spatial audio dataset, is a significant contribution. It provides a valuable resource for future research in text-guided audio spatialization.
3. Semantic-Aware Fusion Module: The SAF module is a key innovation that effectively integrates text and audio features, enhancing the model's ability to generate spatial audio that aligns with textual descriptions.
4. Diffusion Model: The use of a text-guided diffusion model for generating differential waveforms adds robustness and precision to the spatial audio generation process.
5. Comprehensive Evaluation: The paper includes extensive experiments, including quantitative and qualitative analyses, ablation studies, and user studies. These evaluations demonstrate the method's effectiveness and potential for personalized audio spatialization.

**Limitations:**

1. Generalization to Other Datasets: The evaluation is primarily conducted on the TASBench dataset. Additional experiments on other datasets would help validate the generalization capabilities of TAS.
2. Complexity: The model introduces several novel components, which adds to the implementation complexity. This might pose challenges for practical deployment and requires substantial computational resources.
3. Parameter Sensitivity: The performance of the model is sensitive to the choice of hyperparameters, such as the weights in the loss function. This sensitivity requires careful tuning and may affect the model's robustness.
4. Limited Real-World Scenarios: While the TASBench dataset provides a controlled environment for evaluation, real-world scenarios with more complex and diverse audio-visual contexts could provide a more rigorous test of the method's capabilities.

**Suitability:**

3

---

### Official Review · Reviewer_mqzn · 2024-05-23

**Rating:** 6
**Confidence:** 3

**Summary:**

The paper explores the task of converting mono audio into spatial audio by providing the audio and text prompts as input, defined as Text-guided Audio Spatialization (TAS). This task aims to provide a more flexible and personalized approach compared to previous visually guided methods that rely on strict audiovisual data. The audiovisual methods are limited when not strictly time-synchronized or if audio sources are not visible. The paper addresses the task extensively by presenting methods based on machine learning to accomplish the tasks, as well as a dataset for training and evaluation.

**Strengths:**

* The paper presents a novel approach to generating spatial audio.
* The paper contributes a dataset, TASBench Dataset, a fusion module Semantic-Aware Fusion (SAF) module and a text-guided diffusion model. The dataset is used to train and evaluate the model.
* The author(s) present a theoretical concept together with a technical implementation and extensive evaluation, including ablation study and user study.
* Overall the paper is well written and fits the conference topics.

**Limitations:**

* The dataset is based on the FAIR-Play dataset, which includes music instruments only, is recorded in a controlled environment, and has an uneven distribution. Future work could evaluate the presented models on more complex sounds.
* The paper does not provide information about the performance in terms of the speed of generating the spatial audio.

**Suitability:**

3

---

### Official Review · Reviewer_3kfH · 2024-05-25

**Rating:** 3
**Confidence:** 4

**Summary:**

The paper introduces Text-guided Audio Spatialization, a novel task for converting mono audio into spatial audio using text prompts. The authors highlight the limitations of visually guided audio spatialization, which requires strict audiovisual synchronization and object visibility. To address this, they propose a method that leverages text prompts for greater personalization and flexibility. The paper also presents TASBench, the first dataset with dense frame-level annotations for audio spatialization tasks. Experimental results demonstrate the effectiveness of the proposed method, showing it outperforms some visually guided approaches and can generate personalized spatial audio based on text inputs.

**Strengths:**

1. The paper introduces a new task text-guided audio spatialization, which is interensting as it provides an alternative to visually guided methods which have limitations related to object visibility and synchronization.

2. The authors collect a TASBench dataset, which is the first dataset with dense frame-level annotations for audio spatialization, and could potentially serve as a valuable resource for the research community and facilitates further research in this area.

**Limitations:**

1. While the motivation to use text for audio spatialization is good, the current method only adopts very coarse spatial resolutions, annotating sound source locations as left, center, and right. This approach overlooks intermediate positions that could be more precisely captured with visual guidance. The text-based approach needs a better definition of spatial resolutions to avoid limiting the task's contribution and the model's performance.

2. The proposed text-guided spatialization method performs worse than many visually guided counterparts, raising questions about its effectiveness. This disparity may be due to the coarse spatial resolution in the text annotations or other factors. The authors should provide a more detailed justification and analysis to address this concern.

3. The paper would benefit from a case study demonstrating scenarios where visually guided methods fail due to occlusion, interference, or other issues, and where the text-guided method succeeds. This would highlight the advantages of the text-guided approach in specific challenging situations.

**Suitability:**

3

---

### Meta-Review · Area_Chair_nQgj · 2024-06-26

**Recommendation:** Accept (Poster)
**Confidence:** 5

**Metareview:**

The paper proposed a novel framework for text-guided audio spatialization, and contributed a TASBench dataset with dense annotation. Although there are certain limitations such as generalization issue to real-world scenarios, most reviewers agreed that the strength of the paper outweighs the weakness.